# Increasing Adiponectin Signaling by Sub-Chronic AdipoRon Treatment Elicits Antidepressant- and Anxiolytic-Like Effects Independent of Changes in Hippocampal Plasticity

**DOI:** 10.3390/biomedicines11020249

**Published:** 2023-01-18

**Authors:** Douglas A. Formolo, Thomas H. Lee, Jiasui Yu, Kangguang Lin, Gang Chen, Georg S. Kranz, Suk-Yu Yau

**Affiliations:** 1Department of Rehabilitation Sciences, Faculty of Health and Social Sciences, Hong Kong Polytechnic University, Hung Hom, Hong Kong; 2Mental Health Research Center, Hong Kong Polytechnic University, Hung Hom, Hong Kong; 3Department of Affective Disorders, The Affiliated Brain Hospital of Guangzhou Medical University (Guangzhou Huiai Hospital), Guangzhou 510260, China; 4School of Health and Life Sciences, University of Health and Rehabilitation Sciences, Qingdao 266000, China; 5Guangdong-Hong Kong-Macau Institute of CNS Regeneration, Jinan University, Guangzhou 510632, China; 6Interdisciplinary Institute for Personalized Medicine in Brain Disorders, School of Chinese Medicine, Jinan University, Guangzhou 510632, China; 7Departments of Psychiatry & Clinical and Translational Institute of Psychiatric Disorders, First Affiliated Hospital of Jinan University, Guangzhou 510632, China; 8Co-Innovation Center of Neurogeneration, Nantong University, Nantong 226001, China; 9Department of Psychiatry and Psychotherapy, Comprehensive Center for Clinical Neurosciences and Mental Health, Medical University of Vienna, 1090 Vienna, Austria; 10The State Key Laboratory of Brain and Cognitive Sciences, The University of Hong Kong, Pok Fu Lam, Hong Kong

**Keywords:** depression, adiponectin, AdipoRon, antidepressant, anxiolytic, hippocampus, synaptic plasticity

## Abstract

(1) Background: Adiponectin is an adipocyte-secreted hormone that has antidepressant- and anxiolytic-like effects in preclinical studies. Here, we investigated the antidepressant- and anxiolytic-like effects of sub-chronic treatment with AdipoRon, an adiponectin receptor agonist, and its potential linkage to changes in hippocampal adult neurogenesis and synaptic plasticity. (2) Methods: Different cohorts of wild-type C57BL/6J and CamKIIα-Cre male mice were treated with sub-chronic (7 days) AdipoRon, followed by behavioral, molecular, and electrophysiological experiments. (3) Results: 7-day AdipoRon treatment elicited antidepressant- and anxiolytic-like effects but did not affect hippocampal neurogenesis. AdipoRon treatment reduced hippocampal brain-derived neurotrophic factor (BDNF) levels, neuronal activation in the ventral dentate gyrus, and long-term potentiation of the perforant path. The knockdown of N-methyl-D-aspartate (NMDA) receptor subunits GluN2A and GluN2B in the ventral hippocampus did not affect the antidepressant- and anxiolytic-like effects of AdipoRon. (4) Conclusions: Increasing adiponectin signaling through sub-chronic AdipoRon treatment results in antidepressant- and anxiolytic-like effects independent of changes in hippocampal structural and synaptic function.

## 1. Introduction

Depression has a lifetime prevalence of nearly 15% [1] and is the leading cause of disability in the world [2]. Even though antidepressants have been developed over the past 60 years [3,4], advancements were made in terms of increased tolerability rather than efficacy [5,6]. Noteworthy, monoaminergic antidepressants’ efficacy hardly goes beyond 50% [5], with a delayed therapeutic onset ranging from 4 to 8 weeks [5]. Adiponectin is the most abundant plasma protein [7,8] that, when downregulated, has been associated with depression and anxiety-related disorders [9,10,11]. On the other hand, increasing adiponectin signaling in preclinical studies leads to rapid antidepressant- and anxiolytic-like effects [12,13,14,15,16]. Despite that, the potential antidepressant and anxiolytic properties of AdipoRon, a potent adiponectin receptor agonist [17], have been scarcely explored.

Many adipokines can cross the blood–brain barrier (BBB) and regulate mood and cognition [18]. The adiponectin receptors 1 (AdipoR1) and 2 (AdipoR2) are widely expressed in brain regions associated with emotional regulation, including the hippocampus, amygdala, and medial prefrontal cortex (mPFC) [16,19,20]. When peripheral adiponectin levels rise in response to energetic challenges, such as physical exercise [21,22], the low-molecular-weight isoforms of adiponectin cross the BBB [23] and modulate the activity of such brain regions [12,14]. We have previously shown that adiponectin signaling is a key mediator of the neurogenic and antidepressant properties of voluntary wheel running [12,24]. Moreover, our group has also shown that chronic treatment with AdipoRon, a synthetic and orally available adiponectin receptor agonist [17], increases hippocampal neurogenesis in a dose-dependent manner [25] and mimics the effects of physical exercise on restoring hippocampal neuroplasticity and cognitive deficits in diabetic mice [26].

AdipoRon treatment for three weeks prevents the depressive phenotype in association with increased hippocampal neurogenesis and reduced neuroinflammation in corticosterone-treated mice [13]. Additionally, increased adiponectin signaling has been associated with antidepressant- and anxiolytic-like effects in shorter time windows [16,27,28]. Increasing activity of the peroxisome proliferator-activated receptor-γ (PPARγ), an upstream positive regulator of adiponectin synthesis, through systemic administration of rosiglitazone (a PPARγ agonist) increases peripheral adiponectin levels and reduces antidepressant- and anxiety-like behavior in 24 h, which is not observed in adiponectin knockout mice [27]. Moreover, infusion of adiponectin in the ventricles [16] or ventral tegmental area [28] results in rapid antidepressant and anxiolytic responses. Such pre-clinical evidence holds the promise for exogenously increasing adiponectin signaling through its receptor agonist (i.e., AdipoRon) as a more fast-acting and efficient antidepressant and anxiolytic intervention compared to classical serotonergic drugs [9].

The hippocampus has been long implicated in depression pathogenesis [29,30,31]. The N-methyl-D-aspartate (NMDA) receptors play a significant role in hippocampal information processing [32,33,34], and NMDA receptor-associated structural and functional changes in the ventral hippocampal plasticity have been associated with mood impairments [35,36]. Although previous work has linked adiponectin’s therapeutic effects with increased hippocampal neuroplasticity [12,13,24,25,26], new evidence suggests that short-term antidepressant response might not rely on the hippocampus [37,38,39]. Inactivating the ventral hippocampus does not affect the ketamine’s rapid antidepressant properties [37]. Moreover, acute activation of the ventral hippocampus does not elicit antidepressant or anxiolytic responses [37,39]. Common serotonergic antidepressants normally require chronic treatment (14 days) to elicit a therapeutic response in stressed mice [40]. Since increasing adiponectin signaling has been previously shown to elicit a rapid antidepressant response [12,13,14,15,16], we aimed at investigating whether short-term (7-day) AdipoRon treatment would elicit antidepressant- and anxiolytic-like effects in a shorter treatment period than the conventional antidepressants and to test whether these effects would be linked to changes in hippocampal neuroplasticity.

## 2. Materials and Methods

### 2.1. Animals and Experimental Design

All experimental procedures were approved and followed the Animal Subjects Ethics Sub-Committee’s guidelines at The Hong Kong Polytechnic University. Eight-week-old C57BL/6J and CamKIIα-Cre male mice with the same genetic background were group-housed (3–5 per cage) in a holding room with controlled temperature (22 ± 2 °C) and kept under a 12 h light–dark cycle. Mice were fed with standard chow and water ad libitum and could habituate to the housing conditions for 7 days before experiments started. Wild-type animals were randomly assigned to a 7-day treatment protocol with AdipoRon (20 mg/kg, i.p.) or vehicle. The day after the last injection, different cohorts of animals were submitted to behavioral (*n* = 11–12/group), molecular (*n* = 5–10/group), or electrophysiological (*n* = 5/group) experiments (Figure 1a). For knockdown experiments, CamKIIα-Cre mice were randomly assigned to intrahippocampal injection of vehicle (PBS, Control) or adeno-associated viruses (AAV) driving the expression of shRNA targeting the N-methyl-D-aspartate (NMDA) subunits GluN2A or GluN2B (*n* = 7–9/group). After fourteen days of recovery from the surgery, animals were submitted to AdipoRon treatment followed by a battery of behavioral tests (Figure 1b).

### 2.2. Drugs

AdipoRon (MedChemExpress, Princeton, NJ, USA) was dissolved in 0.5% carboxymethylcellulose sodium salt (CMC) (Sigma-Aldrich, St. Loius, MO, USA) at 70 °C in 5% DMSO, as previously performed (26). Animals were treated with AdipoRon (20 mg/kg, i.p.) or vehicle (CMC in 5% DMSO). For adult neurogenesis study, 5-bromo-2’-deoxyuridine (BrdU, Abcam, Cambridge, UK) was dissolved in 0.9% saline solution and administered for three consecutive days (100 mg/kg) before AdipoRon treatment.

### 2.3. Behavioral Tests

Behavioral tests were performed during the afternoon period (1–5 PM) in a procedure room with controlled temperature (22 ± 2 °C) and bright light by an experienced researcher blinded to the treatment conditions. The behavioral analysis was equally performed by a blinded researcher. Animals could habituate to the room conditions for at least 2 h before experiments started.

#### 2.3.1. Sucrose Preference Test

Sucrose preference, a behavior that relies on the rodent’s natural preference for sweetened solutions and of relevance for the screening of antidepressant-like properties [41], was evaluated by the sucrose preference test (SPT) as previously performed [12]. Briefly, mice were habituated to having access to two water bottles for two consecutive days before the test. On the testing day, mice were single housed for 24 h and allowed access to two standard bottles of water, one containing a 2% sucrose solution (*w*/*v*) and another filled with filtered tap water. The positions of the two bottles were swapped after 12 h to avoid any selection bias effect. Liquid consumption was measured by weighing the bottles before and after the test. Sucrose preference was determined by the ratio of sucrose solution intake in comparison to total liquid consumption over the 24 h period.

#### 2.3.2. Novelty-Suppressed Feeding Test

The novelty-suppressed feeding test (NSFT) was used to evaluate the expression of approach-avoidance behaviors that are of relevance for the screening of anxiety-like properties, as previously described [42]. Mice were deprived of food for 16 h, including the overnight period, before the test. A squared arena (L × W × H: 40 × 40 × 30 cm) was used as a novel environment, with the floor covered by 2 cm of wooden bedding and a single food pellet placed on a filter paper at its center. On the testing day, one mouse at a time was placed in the arena randomly facing one of its corners and let explore it freely for up to 10 min, while the latency to feed was recorded by an experienced researcher in real-time. Once the animal bit the food pellet, or after 10 min had passed, the test ended. Wooden bedding was changed, and the arena was cleaned with 70% ethanol solution in between animals.

#### 2.3.3. Light–Dark Box Test

The light–dark box (LDB) test was used as an additional measure of approach-avoidance behavior [43]. The LDB consisted of an apparatus with two chambers of equal dimensions (L × W × H: 20 × 40 × 30 cm) but enclosed either with transparent (light compartment) or black Plexiglas (dark compartment). Both compartments were interconnected by a sliding door. The light compartment was brightly illuminated with a table lamp, while the dark compartment was covered by a removable ceiling. During the test, a mouse was introduced into the black compartment and let freely explore both compartments for 5 min once the interconnecting door was opened [28]. The entire session was video recorded, and the number of visits and the time spent in the light compartment were manually recorded. Entering and exiting compartments were considered when the mouse crossed the door into the opposite compartment with the four paws. The apparatus was thoroughly cleaned with 70% ethanol solution in between animals.

#### 2.3.4. Forced Swim Test

To evaluate behavioral despair, a behavior that has been commonly associated with antidepressant-like properties [44], mice were subjected to the forced swim test (FST) [45]. Mice were individually placed in a transparent cylinder (height: 30 cm; diameter: 15 cm) filled with two-thirds of water (24 ± 2 °C) for 6 min, as previously performed [12]. The session was video recorded and the duration the animal spent immobile in the last 4 min of the test was manually scored by an experienced researcher. Immobility was defined as no limb or body movements except those necessary for keeping the head above water.

### 2.4. Tissue Preparation for Immunostaining

Animals were deeply anesthetized with isoflurane (Zoetis, Parsippany-Troy Hills, NJ, USA) and perfused with 0.9% saline followed by 4% paraformaldehyde (PFA) in 0.01 M phosphate-buffered saline (PBS). The isolated brains were post-fixed with 4% PFA at 4 °C overnight and then stored in 30% sucrose solution until they sank. Coronal sections (1-in-6 series, 30 μm thickness) were prepared using a vibratome (Leica Biosystems, Wetzlar, Germany) and stored in cryoprotectant solution (30% glycerol and 30% ethylene glycol) at 4 °C.

#### 2.4.1. Immunofluorescence Staining

Immunofluorescence double staining of c-Fos, an immediate early gene marker induced by neural activation [46,47], and NeuN, a neuronal marker [48], was performed in free-floating brain sections. Sections were rinsed in 0.01 M PBS, incubated with 10 mM citric acid buffer for 10 min at 95 °C, and rinsed with 2N HCl for 10 min at 37 °C to unmask antigen. After PBS wash, sections were incubated overnight with mouse anti-c-Fos (1:1000, Santa Cruz Biotechnology, Dallas, TX, USA) and rabbit anti-NeuN antibodies (1:1000, Millipore Sigma, Burlington, MA, USA) at room temperature, followed by a 2 h incubation with goat anti-mouse IgG AlexaFluor-568 and goat anti-rabbit IgG AlexaFluor-488 (1:200, Invitrogen, Waltham, MA, USA). After PBS washes, the sections were coverslipped with fluorescent mounting medium (Dako, Santa Clara, CA, USA).

Immunofluorescence double staining of BrdU as a marker of surviving newborn cells [49] and doublecortin (DCX), a marker of immature neurons [50], was performed following the same steps. After antigen retrieval, sections were incubated overnight at room temperature with rat anti-BrdU (1:500, Abcam, Cambridge, UK) and mouse anti-DCX (1:200, Santa Cruz Biotechnology, Dallas, TX, USA), followed by a 3 h incubation with goat anti-rat IgG AlexaFluor-647 (1:200, Invitrogen, Waltham, MA, USA) and goat anti-mouse IgG AlexaFluor-488 (1:200, Abcam, Cambridge, UK) at room temperature. After PBS wash, the sections were coverslipped with fluorescent mounting medium (Dako, Santa Clara, CA, USA).

#### 2.4.2. Immunoperoxidase Staining

Immunoperoxidase staining of Ki-67, a proliferative cellular marker [51], and DCX was performed in free-floating brain sections as previously performed [25]. Briefly, sections were rinsed in 0.01 M PBS and incubated with 10 mM citric acid buffer for 10 min at 95 °C for antigen unmasking. After PBS wash, sections were incubated overnight at room temperature with rabbit anti-Ki-67 (1:1000, Abcam, Cambridge, UK) or mouse anti-DCX (1:200, Santa Cruz Biotechnology, Dallas, TX, USA). After PBS wash, sections were then incubated for 2 h with goat anti-rabbit or goat anti-mouse (1:200; Vector, Newark, CA, USA), followed by a 2 h incubation with Vectastain Elite ABC solution (Vector, Newark, CA, USA) and 5–10 min in 3,3′-diaminobenzidine (DAB) peroxidase kit (Vector, Newark, CA, USA) for visualization. Sections were then mounted onto coated slides, dehydrated with ethanol and xylene baths, and coverslipped with DPX Mountant.

#### 2.4.3. Cell Quantification

Immunofluorescence images were obtained using a confocal laser scanning microscope (Carl Zeiss Microscopy, Jena, Germany) at 200× magnification in intervals of 4 µm in the z-focal plane with 256 × 256 pixels resolution. Images were z-stacked with maximum intensity projection using the ZEN (blue edition) software (Carl Zeiss Microscopy, Jena, Germany). Immunoperoxidase images were visualized at 400× using a microscope (Nikon series Eclipse H600L, Tokyo, Japan).

c-Fos+ cells were counted automatically in ImageJ (National Institutes of Health, Bethesda, MD, USA) using the “Analyze particles” function after background subtraction, as previously described [42]. After confirming co-labeling with NeuN, c-Fos+ cell count per unit area (mm2) in the DG was obtained through the average of 3 to 4 sections in the dorsal (−1.34 to −2.54 mm from Bregma) and the ventral hippocampus (−2.54 to −3.80 mm from Bregma) per animal.

BrdU-, Ki-67-, and DCX-positive cells located in the dentate subgranular zone and granular cell layer were manually counted in 5 to 6 coronal sections per animal along the anteroposterior hippocampal plane (−1.34 to −3.80 mm from Bregma). The total number of positive cells was estimated by multiplying the average by the expected number of 30 µ coronal sections obtained along the referred plane, as adapted from previous works [52,53].

Fifty BrdU+ cells were randomly selected per animal from 5 to 6 coronal sections along the anteroposterior hippocampal plane (−1.34 to −3.80 mm from Bregma), and the ratio of BrdU-labeled cells that co-labeled with DCX was quantified as a marker of neuronal differentiation, as previously performed [12,25].

### 2.5. Hippocampal Synaptoneurosome and Homogenate Preparations

To purify a crude synaptoneurosome, animals were sacrificed approximately 2 h after the last AdipoRon injection and hippocampi were isolated and homogenized in ice-cold Syn-PER synaptic protein extraction reagent (Thermo Fisher Scientific, Waltham, MA, USA), according to the manufacturer’s manual. After sonication, hippocampal homogenates were centrifuged at 1200× *g* at 4 °C for 10 min, and cytosolic fraction was collected and centrifuged at 15,000× *g*, 4 °C, for 30 min. Crude synaptosomal fraction was resuspended in Syn-PER reagent. To extract total proteins from hippocampal homogenate, the hippocampi were homogenized in ice-cold radioimmunoprecipitation assay buffer (Abcam, Cambridge, UK) containing Halt© phosphatase/protease cocktail (Thermo Fisher Scientific, Waltham, MA, USA) and phenylmethanesulfonyl fluoride (Thermo Fisher Scientific, Waltham, MA, USA). Samples were then sonicated for 20 s with a 50% pulse, followed by centrifugation at 14,000× *g* at 4 °C for 30 min. The protein concentrations of hippocampal synaptoneurosome and hippocampal homogenates were quantified by Bradford assay (Bio-Rad Laboratories, Hercules, CA, USA). Samples were stored at −80 °C until use.

#### 2.5.1. Western Blotting

Homogenates were linearized in 1× laemmli buffer (Bio-Rad Laboratories, Hercules, CA, USA) with 5% β-mercaptoethanol (Abcam, Cambridge, UK) at 99 °C for 10 min. Homogenate containing 50 μg protein was loaded in each lane. Proteins were separated by 10% SDS polyacrylamide gel (Bio-Rad Laboratories, Hercules, CA, USA) and transferred to polyvinylidene fluoride (PVDF) membranes (Bio-Rad Laboratories, Hercules, CA, USA). Non-phosphorylated protein targets were blocked by 5% non-fat dry milk powder (Bio-Rad Laboratories, Hercules, CA, USA) and phospho-protein targets were blocked by 5% bovine serum albumin (Sigma-Aldrich, St. Loius, MO, USA) in the Tris-HCl buffer (pH 8.0). After 1 h blocking, membranes were incubated overnight with primary antibodies in blocking solution with 0.05% Tween−20. The following primary antibodies were used: rabbit anti-PSD95 (1:1000, Cell Signaling Technology, Danvers, MA, USA), rabbit anti-GluA1 (1:1000, Cell Signaling Technology, Danvers, MA, USA), rabbit anti-GluN2A (1:1000, Cell Signaling Technology, Danvers, MA, USA), rabbit anti-GluN2B (1:1000, Cell Signaling Technology, Danvers, MA, USA), rabbit anti-phospho-GluN2A1246 (1:1000, Cell Signaling Technology, Danvers, MA, USA), rabbit anti-phospho-GluN2B1472 (1:1000, Cell Signaling Technology, Danvers, MA, USA), rabbit anti-adiponectin R1 (1:1000, Abcam, Cambridge, UK), goat anti-adiponectin R2 (1:1000, Novus Biologicals, Centennial, CO, USA), rabbit anti-α-tubulin (1:1000, Cell Signaling Technology, Danvers, MA, USA), mouse anti-β-actin (1:5000, Invitrogen, Waltham, MA, USA). After washing, membranes were incubated with the corresponding secondary HRP-linked antibodies for 1 h: goat anti-rabbit IgG (1:1000, Cell Signaling Technology, Danvers, MA, USA), goat anti-mouse IgG (1:1000, Cell Signaling Technology, Danvers, MA, USA), and donkey anti-goat IgG (1:1000, Invitrogen, Waltham, MA, USA), and developed by Clarity Western ECL Substrate (Bio-Rad Laboratories, Hercules, CA, USA) for chemiluminescence detection and documented by a transilluminator (Bio-Rad Laboratories, Hercules, CA, USA). Band intensities were subjected to densitometric analysis in the Image Lab software (Bio-Rad Laboratories, Hercules, CA, USA).

#### 2.5.2. Immunoassays for Adiponectin and BDNF Levels

Levels of adiponectin and BDNF in the total hippocampal lysates were determined 22 to 24 h after the last AdipoRon injection by using the commercial ELISA kits, including the mouse adiponectin ELISA kit (AdipoGen Life Sciences, San Diego, CA, USA) for measuring adiponectin levels in the hippocampus and the total BDNF Quantikine ELISA kit (R&D system, Minneapolis, MN, USA) for measuring BDNF levels in the hippocampus according to the manufacturer’s instructions.

### 2.6. Tissue Preparation for Electrophysiology

Brain slices were prepared as previously described [26]. Brains were isolated from adult male mice (8–9 weeks old) 22 to 24 h after the last AdipoRon injection and immersed in a chilled artificial cerebrospinal fluid (ACSF) containing (in mM) 125 NaCl, 25 NaHCO_3_, 3 KCl, 1.25 NaH_2_PO_4_, 10 D-glucose, 1 CaC_l2_, and 6 MgCl_2_, and saturated with 95% O_2_/5% CO_2_. Transverse brain slices (thickness: 350 μm) were obtained using a semi-automatic vibratome (VT1200, Leica Biosystems Inc., Wetzlar, Germany). Slices were gently transferred to an incubation chamber filled with ACSF containing (in mM) 125 NaCl, 25 NaHCO_3_, 3 KCl, 1.25 NaH_2_PO_4_, 10 D-glucose, 2 CaCl_2_, and 1 MgCl_2_, and saturated with 95% O_2_/5% CO_2_. Slices were recovered at 35 °C for a minimum of 1 h before recording.

#### Field Recording

Recordings of field excitatory postsynaptic potentials (fEPSPs) were performed in a multi-electrode array system (Alpha MED Scientific Inc., Japan), as previously described [26]. A recording probe with extracellular electrodes (P515A, Alpha MED Scientific Inc., Osaka, Japan) was used to stimulate granule neurons in the middle molecular layer of the suprapyramidal blade of the hippocampal dentate gyrus. Slices were perfused at a rate of approximately 2 mL/min. fEPSPs were acquired using the MED-A64MD1 and MED-A64HE1S recording amplifiers (Alpha MED Scientific Inc., Osaka, Japan) and Mobius software. For each slice, stimulus intensity (20–30 μA) was adjusted to yield 40–50% of the maximal response slope without population spikes. Baseline fEPSP measurements were obtained by delivering single-pulse stimulation at 15 s interstimulus intervals. After obtaining a steady baseline of at least 20 min, a high-frequency tetanic stimulation (HFS) was used to induce long-term potentiation (LTP) in the presence of 5 μM bicuculline methiodide. The HFS protocol consists of 4 trains of 50 pulses delivered at 100 Hz with a 30 s intertrain interval. After field recording has finished, input/output curves were recorded with increasing stimulation intensity (10 μA steps).

### 2.7. Surgical Procedure for Adeno-Associated Virus Injection

Adeno-associated viruses (AAV2-psico-CMV-mCherry-NR2A-shRNA and AAV2-psico-CMV-mCherry-NR2B-shRNA) were purchased from Virovek (2.1 × 1013 vg/mL, USA). CamKIIα-Cre male mice were anesthetized with a cocktail of ketamine (100 mg/kg) and xylazine (10 mg/kg) and positioned in a stereotaxic apparatus (RWD, Shenzhen, China). Bilateral craniotomies were performed over the hippocampus and a Hamilton syringe (33-gauge needle, Reno, Hamilton, USA) mounted in an automated pump (11 Elite Nanomite, Harvard Apparatus, Holliston, MA, USA) was used to inject a viral bolus or vehicle (PBS, control) of 0.5 µL/site at a rate of 0.8 µL/min in the ventral hippocampus (3 mm posterior, 3.3 mm lateral, and 3.6 mm ventral to Bregma). After injection, the needle was left in place for 5 min before retraction. Animals were treated with Flunixin (2.5 mg/kg, s.c.) for three consecutive days and allowed to recover for 14 days before experimental procedures.

### 2.8. Statistical Analyses

Data were shown as mean ± SEM. The unpaired t-test was employed to compare the effects of AdipoRon versus vehicle treatments in behavioral assessments, histological analyses, fEPSP analyses, western blotting analyses, and ELISA. The Mantel-Cox (log-rank) overall test was employed to examine the survival percentage in the novelty-suppressed feeding test. Two-way ANOVA with Sidak’s post-hoc test was used to evaluate the effects of AdipoRon and stimulus intensity in the input-output response in field electrophysiological recordings. One-way ANOVA with Tukey’s post-hoc test was used to evaluate the effects of AdipoRon on behavioral changes in CamKIIα-specific NR2A and NR2B knockdowns. Statistical analyses were performed in Prism 9.0 software (GraphPad Software, Boston, CA, USA). *p* < 0.05 was considered statistically significant.

## 3. Results

### 3.1. Sub-Chronic AdipoRon Treatment Induced Antidepressant- and Anxiolytic-Like Effects

We first determined whether sub-chronic (7 days) treatment with AdipoRon could modulate the behavioral profile indicative of antidepressant- and anxiolytic-like properties. Treatment significantly modulated behaviors associated with antidepressant-like properties, as observed by a greater preference for sucrose in the SPT (Figure 2a; t_21_ = 3.623, *p* = 0.0016) and reduced immobility time in the FST (Figure 2b; t_21_ = 2.265, *p* = 0.0342) of AdipoRon- compared to vehicle-treated mice. To further explore the potential anxiolytic-like effect of AdipoRon, we conducted NSF and LDB tests. Our data demonstrated that AdipoRon-treated mice had a shorter latency to feed in the NSF test compared to vehicle-treated mice (Figure 2c; chi-square = 3.975, *p* = 0.0462). Besides, in the LDB test, mice receiving AdipoRon treatment spent a longer time (Figure 2d; t_21_ = 3.224, *p* = 0.0041) and visited the light compartment more frequently (Figure 2e; t_21_ = 4.938, *p* < 0.0001) than the vehicle-treated mice. Collectively, our results suggest that AdipoRon treatment for 7 days elicits behavioral changes indicative of antidepressant- and anxiolytic-like effects.

### 3.2. Sub-Chronic AdipoRon Treatment Did Not Change Adult Hippocampal Neurogenesis

Chronic (14 days) AdipoRon treatment has been associated with increased cell proliferation in the hippocampal DG [25,26]. We investigated whether sub-chronic AdipoRon treatment would also change adult hippocampal neurogenesis. Seven-day AdipoRon treatment did not change the number of hippocampal proliferative cells (Ki67+ cells: Figure 3a,d, t_14_ = 1.210, *p* = 0.2462) or immature neurons (DCX+ cells: Figure 3b,e, t_14_ = 0.996, *p* = 0.3359); nor did it affect neural survival (BrdU+ cells: Figure 3c,f, t_14_ = 0.610, *p* = 0.5516), or neuronal differentiation (the ratio of BrdU and DCX co-labeling: Figure 3g, t_14_ = 1.139, *p* = 0.2736). Overall, these data suggest that sub-chronic AdipoRon treatment does not affect hippocampal neurogenesis, suggesting its antidepressant and anxiolytic effects are not linked to adult neurogenesis.

### 3.3. Acute AdipoRon Treatment Suppressed Neuronal Activation in the Ventral Hippocampus

Hippocampal DG neuronal activation has been associated with antidepressant effects [54]. To further investigate whether the AdipoRon antidepressant- and anxiolytic-like effects could be due to increased neuronal activation in the hippocampal DG, we quantified the c-Fos-immunopositive cells (Figure 4a) 2 h after delivering a single dose of AdipoRon. We used a single dose for this experiment because c-Fos is an acute marker of neuronal activation, which peaks around 2 h after stimulation and does not accumulate upon consecutive treatment [47]. Our results showed that acute AdipoRon administration significantly reduced neuronal activation in the ventral DG (Figure 4c; t_10_ = 3.974, *p* = 0.0026), although there was no significant change in the number of c-Fos-positive cells in the dorsal DG (Figure 4b; t_10_ = 1.513, *p* = 0.1612). Moreover, there was a significant reduction in the total number of c-Fos-positive cells in the whole DG (Figure 4d; t_10_ = 2.756, *p* = 0.0203). Such results indicate that a single dose of AdipoRon elicits rapid action in suppressing neuronal activation in the hippocampal DG.

### 3.4. Sub-Chronic AdipoRon Treatment Reduced Protein Levels of Brain-Derived Neurotrophic Factor but Not Synaptic NMDA Receptor Subunits in the Hippocampus

We further investigated whether the level of neurotrophic support in the hippocampus was altered after subchronic AdipoRon treatment. Results showed that AdipoRon treatment reduced the hippocampal BDNF levels (Figure 5a; t_18_ = 2.227, *p* = 0.039). Of note, AdipoRon treatment did not alter the level of adiponectin (Figure 5b; t_18_ = 0.6939; *p* = 0.4966) as well as the expressions of AdipoR1 (Figure 5c,d; t_18_ = 0.6372, *p* = 0.5320) or AdipoR2 (Figure 5c,e; t_18_ = 0.241; *p* = 0.9026) in the hippocampus.

We next tested whether AdipoRon changed the expressions of postsynaptic density protein 95 (PSD-95) and glutamatergic receptor subunits in the hippocampus. Western blotting analysis (Figure 6a) showed no significant change in the expression levels of PSD-95 in the hippocampal synaptoneurosome extraction (Figure 6b; t_8_ = 2.146; *p* = 0.0921). Moreover, the treatment did not alter the expression of NMDA receptor subunits, including the GluA1 subunit of α-amino-3-hydroxy-5-methyl-4-isoxazolepropionic acid (AMPA) receptor (Figure 6c; t_8_ = 0.6580; *p* = 0.5290), the phosphorylation of GluN2A (Figure 6d; t_8_ = 0.987; *p* = 0.3595), and GluN2B (Figure 6e; t_8_ = 1.964; *p* = 0.081). Overall, these results suggest that sub-chronic AdipoRon treatment reduces neurotrophic support without significantly changing NMDA receptor subunits and adiponectin protein expression in the hippocampus. 

### 3.5. Sub-Chronic AdipoRon Treatment Reduced Synaptic Plasticity in the Hippocampal DG

To further confirm whether changes in neurotrophic support would affect hippocampal synaptic plasticity, we examined LTP formation at the perforant path of the hippocampal DG region after sub-chronic AdipoRon treatment. Treatment reduced the HFS-induced potentiation of synaptic responses in the perforant path (Figure 7a), as indicated by a significant reduction in the LTP of AdipoRon-treated mice when compared to vehicle-treated mice (Figure 7b; t_18_ = 2.607, *p* = 0.0173). In agreement with that, basal synaptic transmission efficiency after LTP was different between AdipoRon- and vehicle-treated animals, as depicted by a significant reduction in the input-output response in AdipoRon-treated mice (Figure 7c; effect of AdipoRon: F_1,19_ = 11.97; *p* = 0.0026; effect of stimulus intensity: F_1.403,26.66_ = 162.9, *p* < 0.0001; effect of interaction: F_8,152_ = 12.64, *p* < 0.0001). Post-hoc analysis revealed that such reduction in synaptic transmission efficiency was significant for stimulation intensities above 50 µA (*p* < 0.05). Collectively, these results indicate that sub-chronic AdipoRon treatment inhibits LTP formation, suggesting its action in suppressing hippocampal DG synaptic plasticity in healthy wild-type mice.

### 3.6. Antidepressant- and Anxiolytic-Like Effects of AdipoRon Were Independent of Hippocampal NMDA Receptors

Finally, to confirm whether the antidepressant- and anxiolytic-like effects of sub-chronic AdipoRon treatment were independent of the hippocampal NMDA receptors, we used AAV-shRNA in CamKIIα-Cre mice to specifically knockdown the expression levels of hippocampal GluN2A and GluN2B subunits. Results showed that neither GluN2A nor GluN2B subunit knockdown affected the behavioral profile indicative of antidepressant-like properties of AdipoRon on sucrose preference in the SPT (Figure 8a; F_2,19_ = 0.12, *p* = 0.8844) or behavioral despair in the FST (Figure 8b F_2,20_ = 0.045, *p* = 0.9564) when compared to controls. Furthermore, anxiolytic-like effects of AdipoRon were also not affected by GluN2A or GluN2B knockdown as evidenced by no difference in latency to eat in NSFT (Figure 8c; chi-square = 1.1, *p* = 0.5705), the number of entries (Figure 8d; F_2,19_ = 0.26, *p* = 0.7726) or time spent (Figure 8e; F_2,19_ = 0.70, *p* = 0.5071) in the light compartment of the LDB. The data indicate that the antidepressant- and anxiolytic-like effects of sub-chronic treatment with AdipoRon may be independent of changes in hippocampal plasticity.

## 4. Discussion

Antidepressants have a delayed therapeutic onset [55], which contributes to the disease burden and vulnerability to suicidal behavior [56]. Here, we demonstrated that 7 days of AdipoRon treatment at a dosage of 20 mg/kg effectively modulate behaviors indicative of antidepressant- and anxiolytic-like properties in naïve mice, which is a shorter onset of antidepressant response compared to conventional antidepressants [40], whereas our molecular and electrophysiological results suggest that the hippocampus is not a key structure mediating such responses.

We observed a significant reduction in behaviors associated with depression- and anxiety-like phenotypes after 7 days of AdipoRon administration. Previously, Nicolas and colleagues [13] showed that chronic AdipoRon treatment (21 days) at a dosage of 1 mg/kg prevented corticosterone-induced depression- and anxiety-like behaviors. However, it did not elicit antidepressant- or anxiolytic-like effects in naïve control animals (not exposed to corticosterone). Moreover, we have previously demonstrated that AdipoRon treatment for 14 days at a dosage of 20 mg/kg restored the cognitive deficits and anxiogenic phenotype in an animal model of diabetes. Nonetheless, AdipoRon’s potential therapeutic properties in terms of depression and anxiety remained unclear. Our current data shows that behavioral changes associated with such effects are observable after 7 days of treatment. Noteworthy, this is about half of the time required for classical antidepressants to elicit a therapeutic response [40], suggesting activating adiponergic signaling can elicit a faster antidepressant-like response.

We also observed that the antidepressant- and anxiolytic-like effects of AdipoRon were not associated with increased hippocampal neurogenesis. Specifically, sub-chronic AdipoRon administration did not change the levels of adult hippocampal cell proliferation, survival, or neuronal differentiation, nor did it affect the amount of hippocampal immature neurons. Even though our previous investigations demonstrated that 14 days AdipoRon treatment (20 mg/kg) increased adult hippocampal cell proliferation [25,26], short-term antidepressant responses are unlikely to rely on neurogenic mechanisms due to the time course of the neurogenic cycle [57,58,59]. As demonstrated in vitro, ketamine has no significant effects on gene expression of neurogenesis and proliferative markers [60]. Moreover, it has been shown that the adiponectin-dependent antidepressant and anxiolytic effects of an enriched environment also do not rely on increased neurogenesis [14]. Therefore, our results support the view that compounds with faster antidepressant-like responses can be neurogenesis-independent.

AdipoRon also acutely reduced neuronal activation in the ventral dentate gyrus and sub-chronically reduced hippocampal neurotrophic support (BDNF). Such neurochemical changes were associated with a downregulation of functional plasticity in the perforant path, as observed by reduced LTP in AdipoRon-treated animals. These results agree with the adiponectin’s neurophysiological properties previously reported in the literature [28,61,62]. Of note, adiponectin infusion reduced the intrinsic excitability of dopaminergic neurons in the ventral tegmental area [28], whereas bath-perfusing acute hippocampal slices with AdipoRon reduces dentate granule neurons’ intrinsic excitability [62] and impairs CA1 LTP formation [60].

Our investigation also suggests that the hippocampus may not be a key structure in the antidepressant- and anxiolytic-like effects of sub-chronic AdipoRon treatment. The relevance of the hippocampus in antidepressant effects has also been questioned by others, where it was demonstrated that chemical inactivation of the ventral hippocampus did not interfere with the rapid antidepressant effects of ketamine [37], and acute stimulation of the ventral hippocampus projecting fibers failed to induce antidepressant effects [37,39]. Indeed, activation of the hippocampus (either optogenetically or chemogenetically) is rather associated with increased stress resilience than with antidepressant response [63]. Therefore, our findings support recent evidence that the hippocampus may have a more significant role in depression pathogenesis [29,31] than in antidepressant response [37,38,39].

Hippocampal functional plasticity is substantially dependent on NMDA-associated glutamatergic transmission [32,33,34]. Although we did not observe a significant reduction in the activity of the NMDA receptor subunits GluN2A and GluN2B after subchronic AdipoRon treatment, we conducted knockdown experiments of such sub-units to confirm whether the ventral hippocampal glutamatergic transmission was necessary for the AdipoRon therapeutic effects. Remarkably, AdipoRon antidepressant- and anxiolytic-like effects were independent of ventral hippocampal NMDA receptor-dependent synaptic function, as knockdown of the NMDA receptor subunits did not interfere with the effects of AdipoRon.

Overall, our investigation indicates that increasing adiponectin signaling through its receptor agonist, AdipoRon, changes the behavioral profile indicative of antidepressant- and anxiolytic-like effects, which is already evident after 7 days of treatment. Moreover, we observed that such properties are likely independent of increased hippocampal activity, although the investigation of synaptogenic markers in the hippocampus has not been excluded and warrants further exploration. The involvement of other brain regions requires further investigation. The mPFC expresses high levels of both adiponectin receptors [19,20] and is one of the key brain regions implicated in rapid antidepressant response [42,64]. Moreover, other brain regions that have been associated with rapid antidepressant response, including the amygdala [42], dorsal raphe nucleus [65], and ventral tegmental area [66], express high levels of adiponectin receptors [16,28,67].

## Figures and Tables

**Figure 1 biomedicines-11-00249-f001:**
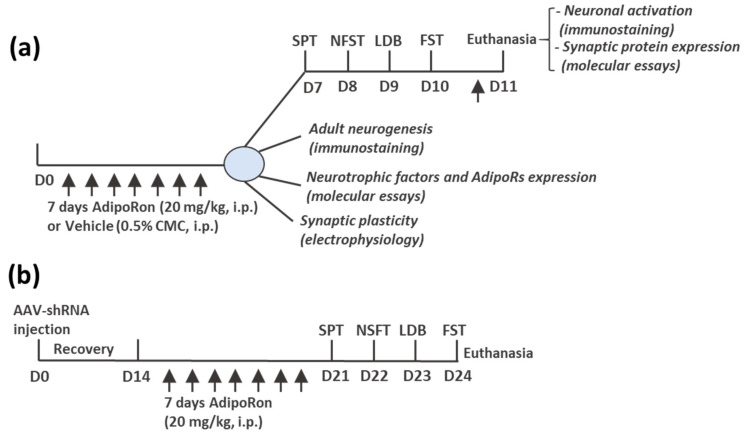
Experimental design. (**a**) Wild-type animals were randomly assigned to a 7-day treatment protocol with AdipoRon (20 mg/kg, i.p.) or vehicle. The day after the last injection, different cohorts of animals were subjected to behavioral, molecular, or neurophysiological experiments. The cohort subjected to behavioral analysis received an AdipoRon injection after the last behavioral test, 2 h before euthanasia. (**b**) CamKIIα-Cre mice were randomly assigned to microsurgery for intrahippocampal injection of PBS (control) or adeno-associated viruses (AAV) driving the expression of shRNA targeting the N-methyl-D-aspartate (NMDA) subunits NR2A or NR2B. Fourteen days after surgery, animals were submitted to AdipoRon treatment followed by a battery of behavioral tests.

**Figure 2 biomedicines-11-00249-f002:**
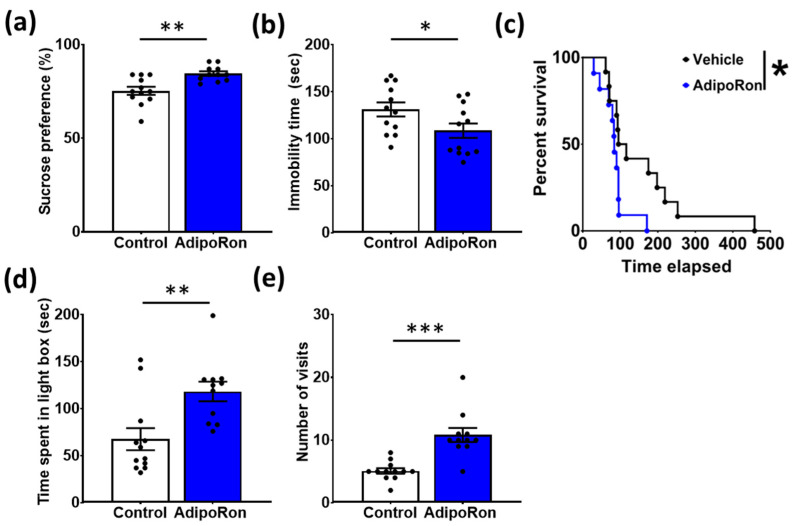
Sub-chronic AdipoRon treatment induced antidepressant- and anxiolytic-like effects. Seven days of AdipoRon treatment significantly (**a**) increased sucrose preference in the SPT and (**b**) reduced immobility time in the FST. Sub-chronic AdipoRon treatment also (**c**) reduced the latency to eat in the NSF test, (**d**) the time spent, and (**e**) the number of visits to the light compartment of the LDB test. Student *t*-test, * *p* < 0.05, ** *p* < 0.01, *** *p* < 0.001. Data presented as mean ± SEM.

**Figure 3 biomedicines-11-00249-f003:**
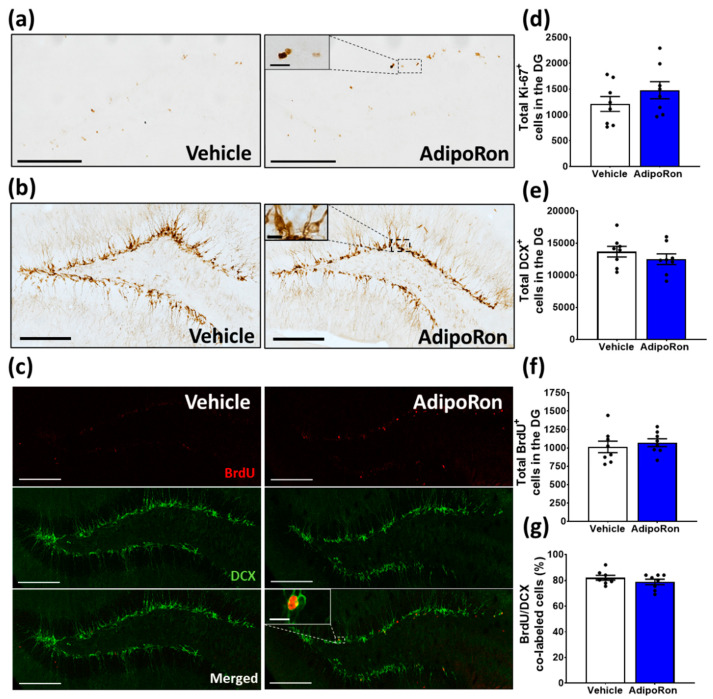
AdipoRon treatment did not change adult hippocampal neurogenesis. Representative images of (**a**) Ki-67+ cells, (**b**) DCX+ cells, and (**c**) BrdU/DCX co-labeled cells in the DG of vehicle- and AdipoRon-treated mice (scale bars, 200 µm in 100×, 10 µm in 400×). Treatment did not significantly change the total expression of (**d**) Ki-67, (**e**) DCX, and (**f**) BrdU-positive cells, nor did it affect (**g**) the percentage of BrdU/DCX co-labeled neurons in the DG. Student *t*-test. Data presented as mean ± SEM.

**Figure 4 biomedicines-11-00249-f004:**
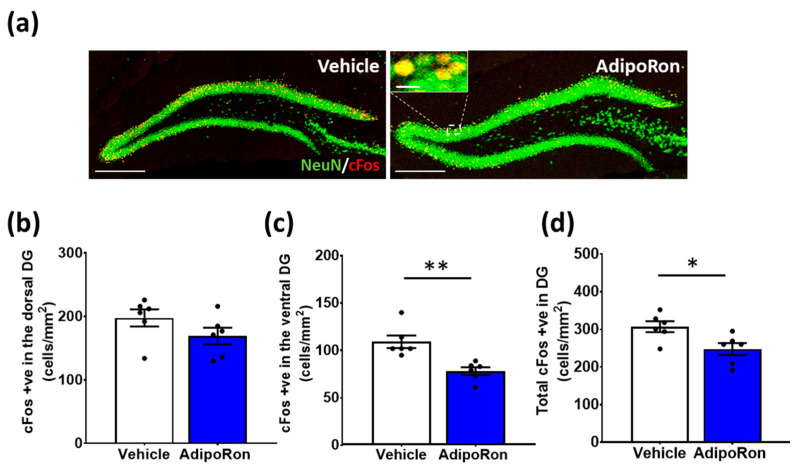
Acute AdipoRon treatment suppressed neuronal activation in the ventral hippocampus. (**a**) Representative images of NeuN/c-Fos co-labeled cells in the DG of vehicle- and AdipoRon-treated mice (scale bars, 200 µm in 100×, 10 µm in 400×). Treatment did not change the expression of c-Fos-positive cells in (**b**) the dorsal DG, but significantly reduced it in (**c**) the ventral DG. (**d**) Combined analyses of the dorsal and ventral DG portions resulted in a significant reduction of the total c-Fos-positive cells in the DG. Student *t*-test, **p* < 0.05, ** *p* < 0.01. Data presented as mean ± SEM.

**Figure 5 biomedicines-11-00249-f005:**
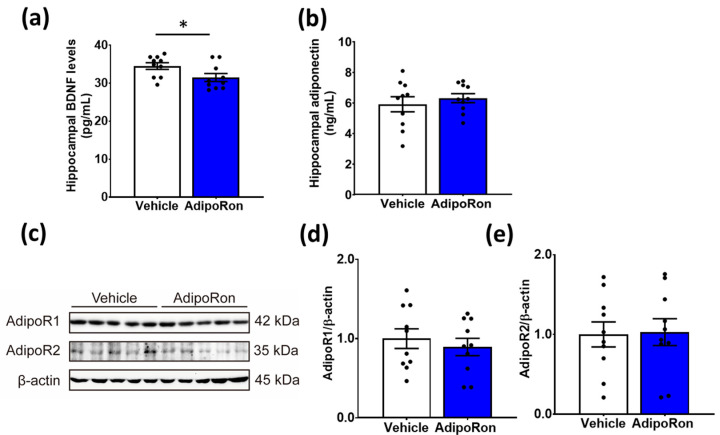
Sub-chronic AdipoRon treatment reduced protein levels of brain-derived neurotrophic factor in the hippocampus. (**a**) Seven-day AdipoRon treatment significantly reduced hippocampal BDNF levels, although changes in the level of (**b**) hippocampal adiponectin. (**c**) Protein expression of (**d**) hippocampal AdipoR1 or (**e**) AdipoR2 was not significantly affected. Student *t*-test, * *p* < 0.05. Data presented as mean ± SEM.

**Figure 6 biomedicines-11-00249-f006:**
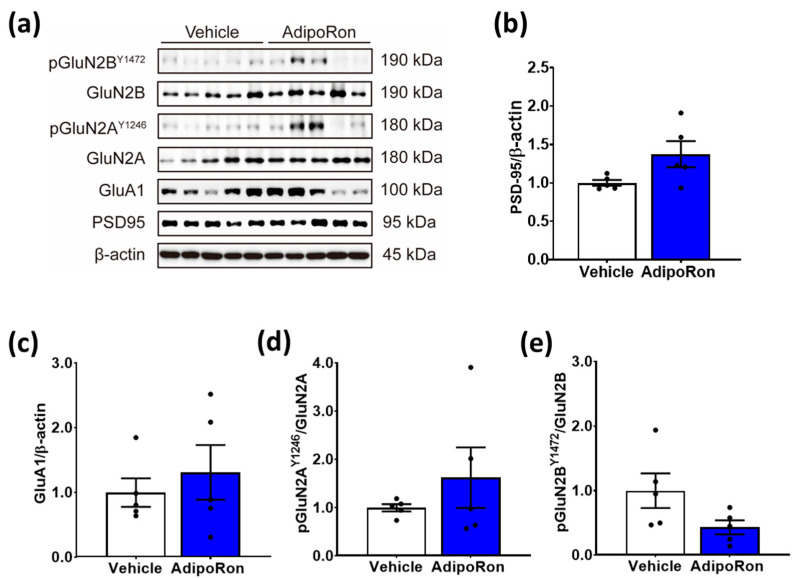
Sub-chronic AdipoRon treatment did not change the expression of synaptic NMDA receptor subunits in the hippocampus. (**a**) Representative images of Western Blot of vehicle- and AdipoRon-treated mice. Seven-day AdipoRon treatment did not significantly change the protein expression of (**b**) PSD-95, (**c**) GluA1 subunit of AMPA receptors, (**d**) GluN2A, or (**e**) GluN2B subunits of NMDA receptors. Student *t*-test. Data presented as mean ± SEM.

**Figure 7 biomedicines-11-00249-f007:**
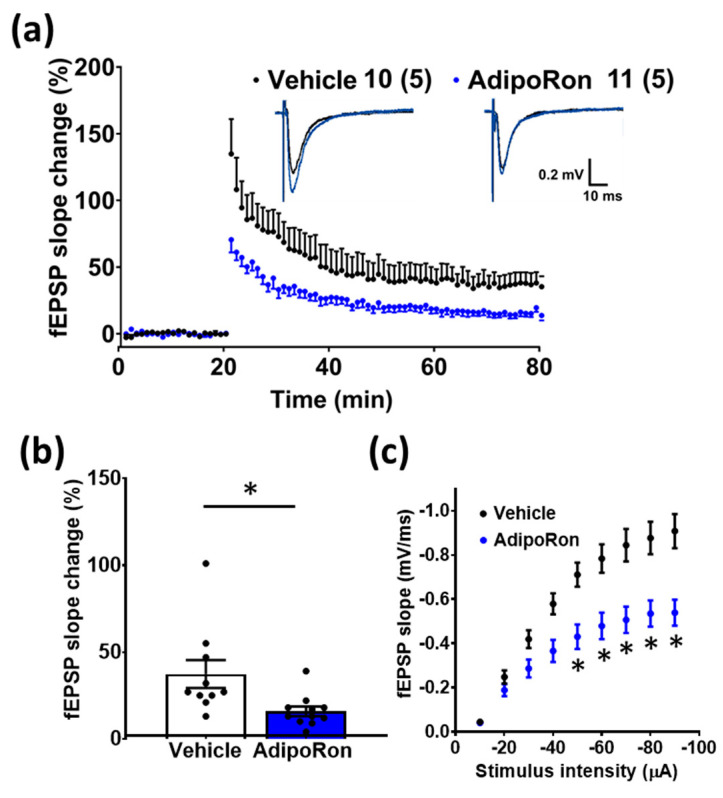
Sub-chronic AdipoRon treatment reduced hippocampal DG synaptic plasticity. Seven-day AdipoRon treatment significantly reduced (**a**) HFS-induced potentiation of synaptic responses in the perforant path, as indicated by (**b**) a significant reduction in the average of the last 5-min LTP of AdipoRon-treated mice (Student *t*-test, * *p* < 0.05). (**c**) Treatment also reduced the basal synaptic transmission efficiency for stimulus intensities above −50 µA (two-way ANOVA with Sidak’s post-hoc test, * *p* < 0.05). Data presented as mean ± SEM.

**Figure 8 biomedicines-11-00249-f008:**
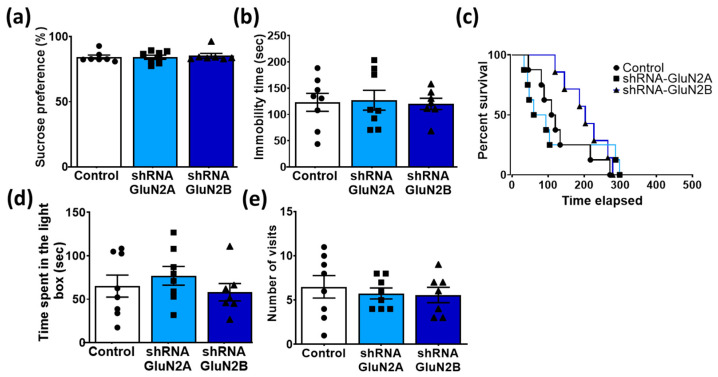
Antidepressant- and anxiolytic-like effects of AdipoRon were independent of hippocampal NMDA receptors. The effects of sub-chronic (7 days) AdipoRon treatment were not significantly affected by knockdown of the NMDA receptor subunit (GluN2A and GluN2B), as no significant alteration was observed in (**a**) the sucrose preference in the SPT, (**b**) immobility time in the FST, (**c**) latency to eat in the NSF test, and (**d**) time spent and (**e**) number of visits to the light compartment of the LDB test, compared with AdipoRon-treated animals with intact NMDA receptor subunits (Control). One-way ANOVA. Data presented as mean ± SEM.

## Data Availability

Data may be made available upon request.

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
