# Peer review of "Increasing Adiponectin Signaling by Sub-Chronic AdipoRon Treatment Elicits Antidepressant- and Anxiolytic-Like Effects Independent of Changes in Hippocampal Plasticity"

_biomedicines, 2023, doi:10.3390/biomedicines11020249_

Round 1

Reviewer 1 Report

The authors detail a series of studies evaluated whether the behavioral effects produced by AdipoRon (20 mg/kg i.p.), an adiponectin receptor agonist, on rodent tests relevant to anxiety and depression were dependent on hippocampal. Neuroplasticity. To that end, adipoRon was administered for 7 days to stress naïve male C57BL/6J and CamKIIα-Cre mice (to facilitate targeted shRNA knockdown of GluN2A or GluN2B NMDA receptors within the hippocampus). Overall, the authors demonstrate a profile of behavioral effects that suggest that this compound will have antidepressant activity when tested clinically. However, these effects are independent of hippocampal neurogenesis, which was not evident with sub chronic treatment. Additionally, studies demonstrated that AdipoRon enhanced the protein expression of post synaptic density 95, but diminished BDNF levels without altering adiponectin levels within the hippocampus. They also demonstrated a reduced number of c-fos positive cells within the dentate gyrus and decreased long term potentiation in the perforant pathway suggesting a possible reduction in neuronal activation and synaptic plasticity within the region.  The authors also demonstrate that there is no effect of hippocampal NMDA receptor knockdown on the behavioral effects of AdipRon, suggesting that it mediates in effects through an alternative mechanism of action.

In the introduction, explicitly state that “Common serotonergic antidepressants normally require chronic treatment (14 days) to elicit a therapeutic response in STRESSED mice”. Typically those classes of medications are behaviorally active in stress naïve rodents immediately.

Rather than stating that the mice have a depressive phenotype, which is somewhat misleading, particularly in this case where animals are stress naïve, just described the behavioral test and the outcomes. Don’t over interpret the data and ascribed human clinical conditions to the mouse. For instance in the SPT, it is just that a sucrose preference test that has relevance to anhedonia, but in these studies where the animals are not exposed to stress and do not exhibit a deficit in sucrose consumption, this is simply a preference task in which AdipoRon gives a positive signal, indicating potential antidepressant-like activity of the compound. This is also true for the FST data, simply state that this test is used to identify compounds with potential antidepressant activity. This is done by using reductions in immobility scores as the primary endpoint. The results and discussion sections should be amended accordingly to remove the depressive phenotype references and do not use the phrase “elicits antidepressant and anxiolytic effects.” These mice do not have these clinical conditions and AdipoRon is not an established anxiolytic or antidepressant in humans.

Was there any training period to establish stable sucrose consumption?

What was the time point post final treatment for the molecular assessments? It is stated that c-fos was assessed 2 h, was this the timepoint for all assessments included e-phys.

Have the authors considered evaluating markers of synaptogenesis, which has been frequently reported with ketamine administration.

Author Response

Dear Reviewer, thank you very much for your valuable comments and suggestions. Please find below a point-by-point response (as well as a world format version attached).

Point 1: In the introduction, explicitly state that “Common serotonergic antidepressants normally require chronic treatment (14 days) to elicit a therapeutic response in STRESSED mice”. Typically those classes of medications are behaviorally active in stress naïve rodents immediately.

Response: Thank you for your observation. Please, refers to the alterations below:

  1. Changes were made according to the request in line 80 of the introduction, which now reads “Common serotonergic antidepressants normally require chronic treatment (14 days) to elicit a therapeutic response in stressed mice”.

Point 2:  Rather than stating that the mice have a depressive phenotype, which is somewhat misleading, particularly in this case where animals are stress naïve, just described the behavioral test and the outcomes. Don’t over interpret the data and ascribed human clinical conditions to the mouse. For instance in the SPT, it is just that a sucrose preference test that has relevance to anhedonia, but in these studies where the animals are not exposed to stress and do not exhibit a deficit in sucrose consumption, this is simply a preference task in which AdipoRon gives a positive signal, indicating potential antidepressant-like activity of the compound. This is also true for the FST data, simply state that this test is used to identify compounds with potential antidepressant activity. This is done by using reductions in immobility scores as the primary endpoint. The results and discussion sections should be amended accordingly to remove the depressive phenotype references and do not use the phrase “elicits antidepressant and anxiolytic effects.” These mice do not have these clinical conditions and AdipoRon is not an established anxiolytic or antidepressant in humans.

Response: Thank you very much for your observation. The following changes have been made.

  1. The text has been adjusted to reduce overinterpretation and attribution of human characteristics to the rodent’s behavior. In this regard, instead of referring to the results as “antidepressant and anxiolytic effects”, they are now described as “behavioral changes that are indicative of antidepressant- and anxiolytic-like properties”. Such changes can be tracked in the following sessions:
    1. Results, session 3.1. (lines 334-337), “We first determined whether sub-chronic (7 days) treatment with AdipoRon could modulate the behavioral profile indicative of antidepressant- and anxiolytic-like properties. Treatment significantly modulated behaviors associated with antidepressant-like properties […]”. In the same session (lines 345 to 347), “Collectively, our results suggest that AdipoRon treatment for 7 days elicits behavioral changes indicative of antidepressant- and anxiolytic-like effects.”. In agreement with that, the heading of this session has been adjusted to “3.1. Sub-chronic AdipoRon treatment induced antidepressant- and anxiolytic-like effects”.
    2. Results, session 3.6 (449-451), “Results showed that neither GluN2A nor GluN2B subunit knockdown affected the behavioral profile indicative of antidepressant-like properties of AdipoRon […]”. Likewise, the heading of this session has been changed to “3.6. Antidepressant- and anxiolytic-like effects of AdipoRon were independent of hippocampal NMDA receptors”.
  • Discussion, session 4. (lines 469-472), “Here, we demonstrated that 7 days of AdipoRon treatment at a dosage of 20 mg/kg effectively modulate behaviors indicative of antidepressant- and anxiolytic-like properties in naïve mice […].”. Similar changes have been made in lines 474-476; 484; and 531.
  1. In order to guarantee the coherence across the manuscript, the following changes have also been made in the Methods sessions corresponding to each of the investigated behaviors.
    1. Session 2.2.1 Sucrose Preference Tests, the text has been changed and now reads “Sucrose preference, a behavior that relies on the rodent’s natural preference for sweetened solutions and of relevance for the screening of antidepressant-like properties (41), was evaluated by the sucrose preference test (SPT) as previously per-formed (12).” (lines 133-135).
    2. Session 2.2.2 Novelty-Suppressed Feeding Test, the text has been changed and now reads “The novelty-suppressed feeding test (NSFT) was used to evaluate the expression of approach-avoidance behaviors that are of relevance for the screening of anxiety-like properties, as previously described (42).” (lines 145-146).
  • Session 2.2.3 Light-Dark box Test, the text has been changed and now reads “The Light-dark box test (LDB) was used as an additional measure of approach-avoidance behavior (43).” (lines 156-157).
  1. Session 2.2.4 Forced swim Test, the text has been changed and now reads “To evaluate behavioral despair, a behavior that has been commonly associated with antidepressant-like properties (44), mice were subjected to the forced swim test (FST) (45).” (lines 169-171).
  1. Likewise, changes have also been made across the Abstract (lines 26; 27; 31; 36; 37), Introduction (lines 50; 68; 91), Results (lines 340; 349; 375; 446; 453; 457; 460), and Discussion (lines 476; 479; 487; 488; 499; 511; 526; 532) to include the qualifier “-like” (e.g., antidepressant-like and anxiolytic-lie) whenever referring to potential therapeutic properties associated with AdipoRon. Of note, such adjustment also included the manuscript title, that now reads “Increasing Adiponectin Signaling by Sub-chronic AdipoRon Treatment Elicits Antidepressant- and Anxiolytic-Like Effects Independent of Changes in Hippocampal Plasticity”

Point 3: Was there any training period to establish stable sucrose consumption?

Response: Thank you for question.

  1. There was no habituation to sucrose solution, as routinely done in our lab (Yau et al., 2022, PMID 34418475). Mice remained group-housed before tests, but had access to two water bottles for 2 days before the SPT. On the day of SPT, mice were single housed with one water bottle and one sucrose bottle. This information has been added to the respective method description as follows (lines 136-138) “Briefly, mice were habituated to having access to two water bottles for two consecutive days before the test. On the testing day, mice were single housed for 24 h and allowed access to two standard bottles of water, one containing a 2% sucrose solution (w/v) and another filled with filtered tap water.

Point 4: What was the time point post final treatment for the molecular assessments? It is stated that c-fos was assessed 2 h, was this the timepoint for all assessments included e-phys.

Response: Thank you for your question. 

  1. To examine synaptic protein expression, we have adopted a staggering schedule in a way that the last injection was performed approximately 2 hours before harvesting fresh hippocampal tissues, corresponding to the group of mice perfused for the immunostaining of cFos/NeuN. This information has been included in the appropriate method description (240-241), “To purify a crude synaptoneurosome, animals were sacrificed approximately 2 hours after the last AdipoRon injection and hippocampi were isolated and homogenized in ice-cold Syn-PER synaptic protein extraction reagent (Thermo Fisher Scientific, MA, USA), according to the manufacturer’s manual.”
  2. To examine neurotrophic factor and AdipoR expression, fresh tissues were harvested 22-24 h after the last injection, which has been included in the method description (line 281), “Levels of adiponectin and BDNF in the total hippocampus lysates were determined 22 to 24 h after the last AdipoRon injection by using the commercial ELISA kits”
  3. To perform electrophysiological recordings, we also adopted a staggering schedule. The 7-day continuous adiporon or vehicle administration has been scheduled in a way that the recording was performed around 22-24 h after the last injection. Likewise, this information has been added to the manuscript (line 288), “Brain slices were prepared as previously described (26). Brains were isolated from adult male mice (8-9 weeks old) 22 to 24 h after the last AdipoRon injection and immersed in a chilled artificial cerebrospinal fluid (ACSF)”.

Point 5: Have the authors considered evaluating markers of synaptogenesis, which has been frequently reported with ketamine administration.

Response: Thank you very much for your observation. We agree that this would be an interesting and complementary information to our findings. Unfortunately, this hasn’t been included in our experimental plan. Nonetheless, due to its relevance, we have included this observation in lines 540-542, “Moreover, we observed that such therapeutic properties are likely independent of in-creased hippocampal activity, although the investigation of synaptogenic markers in the hippocampus hasn’t been excluded and warrants further exploration.”

Reviewer 2 Report

Authors should better explain the criterion for choosing a single dose of treatment (lines 370-376).

Did the authors observe any alteration in cumulative food intake after AdipoRon treatment?

The alteration of BDNF could also suggest a potential modulation of neural circuitries involved in feeding control.

Reference section should be edited according to journal guidelines.

Author Response

Dear Reviewer, thank you very much for your valuable comments and suggestions. Please find below a point-by-point response (as well as a world format version attached).

Point 1: Authors should better explain the criterion for choosing a single dose of treatment (lines 370-376).

Response: Thank you for your comments. The following explanation has been added to the text (lines 380-382), “To further investigate whether the AdipoRon antidepressant- and anxiolytic-like effects could be due to increased neuronal activation in the hippocampal DG, we quantified the c-Fos-immunopositive cells (Fig. 4a) 2 hours after delivering a single dose of AdipoRon. We used a single dose for such experiment because c-Fos is an acute marker of neuronal activation, which peaks around 2 h after stimulation and does not accumulate upon consecutive treatment (47).”

Point 2: Did the authors observe any alteration in cumulative food intake after AdipoRon treatment? The alteration of BDNF could also suggest a potential modulation of neural circuitries involved in feeding control.

Response: Thank you for this observation. Indeed, although with contradicting findings, modulation of central adiponectin signaling (with emphasis in hypothalamic signaling systems) has been associated with changes in food intake. However, we have not measured cumulative food intake in our experimental design, mostly because we wanted to avoid the potential stress arising from social isolation in case the animals were single housed for food intake monitoring.  

Point 3: Reference section should be edited according to journal guidelines.

Response: Thank you for this observation. Reference section was adjusted accordingly.

Reviewer 3 Report

Dear Authors, 

your previous publications indicate expertise in the topic of the manuscript (adinonectin activity in depression and anxiety) and in the experimental protocols used. As indicated in the discussion, it should be of great interest to evaluate the role of other brain areas (cortex and amygdala) involved in the earlier antidepressant and anxiolytic effects of adipoRon than current therapeutic treatments.

Minor points to be clarified are as follows.

1.       Include acute treatment (D11) in the legend of Figure 1 (panel a).

2.       Clarify the treatment duration (acute or subchronic) in the synaptic protein expression method and related results shown in Figure 6a.

3.       Include "suchronic" in the title of section 3.2.

Author Response

Dear Reviewer, thank you very much for your valuable comments and suggestions. Please find below a point-by-point response (as well as a world format version attached).

Point 1: Include acute treatment (D11) in the legend of Figure 1 (panel a).

Response: Thank you for your observation. This information has been added accordingly (lines 114-116), “The day after the last injection, different cohorts of animals were subjected to behavioral, molecular, or neurophysiological experiments. The cohort subjected to behavioral analysis received an AdipoRon injection after the last behavioral test, 2 h before euthanasia.”

Point 2: Clarify the treatment duration (acute or subchronic) in the synaptic protein expression method and related results shown in Figure 6a.

Response: Thank you for your suggestion. The following information has been included in the respective methods.

  1. To examine synaptic protein expression, this information has been included in the appropriate method description (240-241), “To purify a crude synaptoneurosome, animals were sacrificed approximately 2 hours after the last AdipoRon injection and hippocampi were isolated and homogenized in ice-cold Syn-PER synaptic protein extraction reagent (Thermo Fisher Scientific, MA, USA), according to the manufacturer’s manual.”
  2. To examine neurotrophic factor and AdipoR expression (line 281), “Levels of adiponectin and BDNF in the total hippocampus lysates were determined 22 to 24 h after the last AdipoRon injection by using the commercial ELISA kits”
  3. To perform electrophysiological recordings, (line 288), “Brain slices were prepared as previously described (26). Brains were isolated from adult male mice (8-9 weeks old) 22 to 24 h after the last AdipoRon injection and immersed in a chilled artificial cerebrospinal fluid (ACSF)”.

Point 3: Include "suchronic" in the title of section 3.2.

Response: Thank you for your suggestion. This information has been included accordingly (line 359)